# Risankizumab: Daily Practice Experience of High Need Patients

**DOI:** 10.3390/biomedicines11061769

**Published:** 2023-06-20

**Authors:** Alexandra M. G. Brunasso, Martina Burlando, Fabrizio Amoruso, Luisa Arancio, Giovanna Malara, Raffaella Manzo, Maria Antonia Montesu, Giacomo Caldarola

**Affiliations:** 1Department of Internal Medicine, Villa Scassi Hospital ASL3, 16100 Genoa, Italy; 2Department of Dermatology, Dipartimento di Scienze Della Salute, DISSAL, University of Genoa, 16100 Genoa, Italy; martina.burlando@unige.it; 3IRCCS Opsedale Policlinico San Martino, 16100 Genoa, Italy; 4Dermatology Unit, Azienda Ospedaliera di Cosenza, 87100 Cosenza, Italy; fabrizio_amoruso@hotmail.com; 5Unit of Dermatology, Fondazione IRCCS Cà Granda Ospedale Maggiore Policlinico, 20019 Milan, Italy; lmhorange@hotmail.com; 6Unità Operativa Complessa di Dermatologia, Azienda Ospedaliero-Universitaria, 89121 Reggio Calabria, Italy; giovanna.malara@ospedalerc.it; 7U.O.C. Dermatologia, ASL Salerno Ospedale Tortora Pagani, 84121 Salerno, Italy; lellamanzo@alice.it; 8Department of Medicine, Surgery and Pharmacy, University of Sassari, 07100 Sassari, Italy; mmontesu@uniss.it; 9UOC di Dermatologia, Dipartimento di Scienze Mediche e Chirurgiche, Fondazione Policlinico Universitario A. Gemelli—IRCCS, 00118 Rome, Italy; giacomo.caldarola@unicatt.it; 10Dermatologia, Dipartimento di Medicina e Chirurgia Traslazionale, Università Cattolica del Sacro Cuore, 00118 Rome, Italy

**Keywords:** psoriasis, risankizumab, monoclonal antibody

## Abstract

Psoriasis is a chronic inflammatory disease which affects 29.5 million people worldwide and it can negatively impact quality of life, especially when it affects a special localization, such as nails, face, palms and soles, or intertriginous regions. Risankizumab is a humanized monoclonal antibody which targets the p19 subunit of interleukin-23 and it is currently licensed also as systemic therapy for moderate to severe plaque psoriasis. Here, we present eight cases of patients with moderate to severe psoriasis treated with risankizumab with a significant efficacy in the remission of the disease. Our cases represent a real-world clinical setting and provide a valuable adjunct to results obtained in the selected patients usually included in controlled clinical trials. In our cases, risankizumab rapidly improved clinical manifestations and relieved symptoms in patients with moderate to severe psoriasis, regardless of the presence of comorbidities or the location of the plaques in special sites, and without any safety concerns.

## 1. Introduction

Psoriasis is a chronic inflammatory disease which affects 29.5 million people worldwide [1], with a prevalence of 1.8–3.1% among the Italian population [2].

Psoriasis can negatively impact quality of life, especially when it affects special localization, such as nails, face, palms and soles, or intertriginous regions [3]. For this reason, psoriasis affecting these special sites requires effective treatment [3].

High needs patients have been defined in the present case series as adult cases with moderate to severe plaque psoriasis who were either not controlled by, were intolerant to, or had contraindications to at least two currently available systemic therapies (e.g., photochemotherapy, cyclosporine, methotrexate, oral retinoids, fumaric acid esters, and biologicals) or require a fast control of the disease due to high severity.

Risankizumab is a humanized monoclonal antibody and it has been licensed also as systemic therapy for moderate to severe plaque psoriasis that targets the p19 subunit of interleukin-23 (IL-23). In 2019, risankizumab was approved by the Food and Drug Administration (FDA) and European Medical Agency (EMA) for the treatment of moderate to severe plaque psoriasis [4].

By blocking IL-23 from binding to its receptor, risankizumab inhibits IL-23-dependent cell signaling and release of proinflammatory cytokines [5,6].

Risankizumab exhibits linear pharmacokinetics when administered intravenously (0.01 mg/kg–1200 mg) or subcutaneously (0.25 mg/kg–300 mg), with a half-life of approximately 28 days.

The efficacy and safety of risankizumab was assessed in 2109 subjects with moderate to severe plaque psoriasis in four multicenter, randomized, double-blind studies (UltIMMa-1, UltIMMa-2, IMMHANCE, and IMMVENT). UltIMMa-1 and UltIMMa-2 were replicate phase 3, randomized, double-blind, placebo-controlled and active comparator-controlled trials which involved 997 subjects (506 in UltIMMa-1 and 491 in UltiMMa-2) worldwide. In both trials, patients were randomized to receive 150 mg risankizumab, 45 mg or 90 mg ustekinumab, or placebo. Results showed superior efficacy of risankizumab compared to both placebo and ustekinumab in the treatment of moderate to severe plaque psoriasis, without any unexpected safety findings [7].

The IMMhance study was a 2-year, phase 3, multinational, double-blind placebo-controlled trial with randomized withdrawal and retreatment involving 507 subjects and comparing risankizumab, 150 mg, with placebo. Results showed that a significantly greater proportion of patients treated with risankizumab vs. placebo achieved a treatment response at week 16 and with long-term continuous risankizumab compared with withdrawal to placebo at 52 and 104 weeks [8].

IMMvent was a phase 3, randomized, double-blind, multicenter active-comparator-controlled trial, which involved 605 patients randomly assigned to receive either risankizumab (*n* = 301, 50%) or adalimumab (*n* = 304, 50%). In this trial, risankizumab showed significantly higher efficacy compared to adalimumab in patients with moderate to severe plaque psoriasis, without any safety warning [9].

Here, we present eight cases of high need psoriasis patients treated with risankizumab with a significant efficacy in the remission of the disease. National and international guidelines suggest using biological therapy after failure of conventional systemic therapy but clear recommendations on which biological to use as first-line therapy are missing. Registrative clinical trials are not representative of real life, which is much more heterogenous. Our cases represent a daily practice setting, where we can find comorbidities, lack of adherence, and subtypes of disease that are normally excluded from clinical trials.

## 2. Case Series

All patients provided written informed consent to receive risankizumab and for their details and images to be published as a case study, subject to non-identification. Ethics committee or institutional review board approval was not necessary for the individual cases reported in this series because each case reflects a retrospective description of clinical findings.

For a summary of patient’s characteristics, previous therapies, and reason for choosing therapy, see Table 1.

### 2.1. Subsection

#### 2.1.1. Psoriatic Patient with Cardiovascular Comorbidities

A 68-year-old male patient with psoriasis was seen in September 2019. He was a former strong smoker, regularly consumed alcohol and reported a family history of psoriasis (mother and sister). He had a history of arterial hypertension since the age of 35 years, pulmonary tuberculosis in 2004, implantation of pacemaker in 2007, myocardial infarction in 2012. He was treated with antihypertensive, cardiological, and anticoagulant therapy. He had received only topical therapies (corticosteroids and vitamin D derivates) with inconsistent response due to lack of adherence.

Physical examination showed extensive infiltrated erythematous-squamous plaques spread over the entire skin area, with marked involvement of the limbs, as reported in Figure 1A,B.

The Psoriasis Area and Severity Index (PASI) score was 45 and the affected body surface area (BSA) was 80% and Dermatology Life Quality Index (DLQI) was 28, showing a significant impact on quality of life.

No psoriatic onychopathy or joint involvement was detected. Based on the patient’s clinical history, the possibility of using traditional systemic drugs was excluded and, in accordance with the pulmonary and cardiological evaluations, the use of an anti-interleukin was considered. Considering the lack of adherence and the presence of various and severe comorbidities, we chose risankizumab therapy which was started at 150 mg s.c on October 2019.

Figure 2, related to the follow-up visit performed in December 2019, at the end of the induction phase, highlights the complete remission of the skin manifestations with achievement of PASI 100 and a DLQI value of 0.

During the COVID 19 pandemic, the patient continued therapy, with regular dosing every 12 weeks, with no side effects.

Figure 3, from September 2021, demonstrates maintenance of PASI 100 after 24 months.

Key points

This clinical case highlights the rapid and complete response to the therapy of severe psoriasis, without any side effect in a severe comorbid patient and in a difficult pandemic era.It is worth emphasizing the excellent therapeutic adherence by the patient, correlated to the administration every 3 months that facilitated the compliance.

#### 2.1.2. Psoriatic Female Obese Patient

A 45-year-old, obese female patient (BMI 34.6) with psoriasis (PASI 27) came to our observation. The patient has suffered from psoriasis since 2016 and has no family history of it. No other comorbidities had been reported. The patients referred previous treatments with cyclosporine and ustekinumab. From January 2016 to June 2016, she took cyclosporine at a dosage of 250 mg twice a day (BID), then scaled down and stopped because of guideline suggestions (maximum of three to six months of therapy). We treated her with ustekinumab, as it seemed to be the most suitable drug for a young girl, due to its dosage every 12 weeks. After the first injection of ustekinumab, the patient had a polymorphous erythema, which led to the treatment discontinuation and the switch back to cyclosporin.

After the discontinuation of cyclosporin, we started a treatment with secukinumab in November 2016, which significantly improved the symptoms, as reported in Figure 4.

After the improvement, secukinumab lost efficacy in June 2017, as reported in Figure 5.

After the treatment failure, the patient had been treated with ixekizumab since June 2017 and for two years, leading to a new improvement of the symptoms, with the achievement of a PASI 0 (Figure 6).

Afterwards, because of secondary inefficacy of ixekizumab, a treatment with risankizumab was started in December 2019, leading to the complete remission of the symptoms. The remission started after 4 weeks with the achievement of a PASI 7, was complete after 12 weeks, and is still maintained at week 164 (Figure 7).

Key points

Obese patients do not do well with most drugs. IL-23 (interleukin-23) blockers, and in the case of UltIMMa-1 and -2 studies specifically, risankizumab is highly effective even in obese patients [7].A very complex patient, obese, who experienced an adverse reaction with the first biologic drug. Subsequent biologics were effective for a limited time-period, and then the patient always experienced significant worsening. With risankizumab, the remission started after 4 weeks with the achievement of PASI 7 and was completed after 12 weeks and maintained until last follow-up visit at week 164.

#### 2.1.3. Psoriatic Patient with Onychopaty and Palmoplantar Involvement

We first observed a 67 year-old male on October 2017 in our outpatient clinic for psoriasis with a severe plaque psoriasis (PASI: 26, onychopathy and palmoplantar involvement). The patient was previously treated with cyclosporine, systemic steroids, and itraconazole. Current treatment for hypertension, hypercholesterolemia and previous heart infarct includes atenolol, cardio aspirin, and atorvastatin. Laboratory analysis revealed previously resolved hepatitis B infection with normal liver function. Since January 2018, the patient was treated at that time with secukinumab at 300 mg standard dose for 19 months when secondary efficacy failure was observed with PASI score increase to 23. Due to the severity of skin lesions with fissures on the lower extremities, in August 2019, the patient was admitted to the hospital for a complicated erysipela that required endovenous ceftriaxone and monitoring. One month later, the patient presented with a PASI score of 35 and severe palmoplantar and nail involvement and brodalumab therapy was initiated but after 16 weeks of therapy only a minimal improvement was recorded (PASI score: 26) and in January 2020, risankizumab therapy was initiated at 150 mg s.c. day-0, after 4 weeks and every 12 weeks thereafter. After only 4 weeks, PASI-90 response was achieved and at week 16, complete clearance was recorded.

After 18 months of continuous therapy, the patient remains clear and compliant to therapy.

The improvement is reported in Figure 8.

Key points

In clinical trials, risankizumab showed a mean reduction in the Nail Psoriasis Severity Index (NAPSI) score of 40% at week 12, which rose to 73% at week 48, compared to 18% decrease in NAPSI score in the ustekinumab group [10].The fast onset of action of risankizumab in a patient treated previously with anti-IL-17 was recorded.Palmoplantar response was recorded in parallel to plaque and nail psoriasis improvement, such a rapid improvement of all districts is not always seen during other therapies such as anti-TNF-alpha therapies [11].

#### 2.1.4. Patient with Palmoplantar Psoriasis and Hepatitis B

A 50-year-old male patient with palmoplantar psoriasis since 18 years old and affected by hepatitis B since 42 years old (treated with tonofovir) came to our observation.

Before starting the treatment with risankizumab, the patient experienced numerous treatment failures with acitretin (several cycles in monotherapy from 2009 to 2012, and subsequently in combination with biological therapy), etanercept (from December 2012 to June 2015, adalimumab (June 2015 to October 2015), ustekinumab (from October 2015 to July 2016), secukinumab (July 2016 to January 2018), and ixekizumab (January 2018 to July 2020).

In July 2020, we started the therapy with risankizumab, which led to the gradual and complete remission of the symptoms in three months, as reported in Figure 9.

In November 2020, the follow-up visit confirmed the complete remission of the symptoms without any reported adverse event. No reactivation of HBV infection had been experienced.

Key points

The psoriasic treatment of patients affected by hepatitis B represents a real challenging problem, since no guidelines are available [12].Risankizumab has been found to be effective and safe in a patient with a main palmoplantar involvement. This is worthy of note because this kind of patient is not enrolled in clinical trials.

#### 2.1.5. Young Psoriatic Patient Started on Risankizumab

A 23-year-old female patient with psoriasis vulgaris for about 5 years came to our observation. The patient had had previous treatments with topical steroids, emollients, phototherapy and in the last year with cyclosporine, which led to a partial and transient benefit.

Objective examination revealed diffuse erythematous-squamous patches on trunk and limbs, particularly infiltrated on legs (as reported in Figure 10), PASI 32, and DLQI 25. The patient reported intense itching and great discomfort in her daily life because the clinical skin manifestations affected her way of dressing, playing sports, her relationship with other people, and her self-esteem, negatively impacting her quality of life.

The patient started therapy with risankizumab 150 mg s.c. in January 2020.

After 4 weeks, during a clinical re-evaluation, a significant improvement was observed, with a PASI 12 and a DLQI 10.

After 12 weeks, before the third administration of the drug, complete resolution of the clinical picture was observed, with a PASI 0 and a DLQI 0. The long-term effects are still appreciable, as reported in Figure 11.

Key points

Adherence tends to be harder to maintain in younger patients, less frequent injections might improve treatment continuation.The efficacy of risankizumab allowed the patient to reach a PASI 0 (which was confirmed after 18 months) and to significantly improved her quality of life.

#### 2.1.6. Scalp Psoriasis

Our clinical case concerns a 38-year-old woman suffering from psoriasis since the age of 21. She was basically in good general health conditions except for an undiagnosed difficulty in conceiving: in fact, the patient had had to resort to in vitro fertilization (IVF) to achieve a full-term pregnancy. In the past, in 2005–2006, she had been treated with cyclosporine, which was discontinued due to intolerance that resulted in gastritis, hair loss, and increased transaminases. Since then, the patient had managed her psoriasis with topical corticosteroid therapy alone, combined with vitamin D derivatives.

The patient came to our attention in January 2020 due to a severe worsening of psoriasis. The psoriasis was widespread and bothersome. The patient suffered mainly from localization to the scalp because of a severe inflammatory state with significant scaling and itching, with an evident strong emotional impact and a strong discomfort in daily life. She presented a PASI 22.6 and DLQI 20. In January 2020, she took the first dose of risankizumab.

After one month, the patient was very satisfied with the result and the discomfort she experienced every day had almost disappeared as demonstrated by PASI (0.5) and DLQI (1) indices (Figure 12). The scalp was almost in remission with a modest residual psoriasis on the nape of the neck and in the retroauricular area, almost asymptomatic.

Key points

Numerous clinical trials demonstrated that risankizumab was effective to reduce the Psoriasis Scalp Severity Index up to 94%, with a higher rate compared to ustekinumab [13].Clinical experience has shown the high efficacy of risankizumab in difficult sites, particularly on the scalp.

#### 2.1.7. Patient with Sensitive and Hard-to-Treat Areas Involvement

A 46-year-old man presented to our examination after many unsuccessful therapeutic attempts. He did not present any comorbidity except for a slight hypercholesterolemia. He stated that he had suffered from psoriasis since the age of 35 years. He had already performed numerous therapeutic treatments in the past methotrexate, cyclosporine, acitretin, PUVA therapy, and, after a break of about a decade, again adalimumab and secukinumab. These treatments were carried out over the years by a number of centers and cannot be dated, since they appeared from the ananmnestic collection.

The cutaneous clinical situation was rather compromised and was characterized by the presence of diffuse plaque psoriasis with particular localization on visible areas such as the face and on difficult-to-treat areas, such as the scalp and genital area. Moreover, he presented a PASI score of 18.2, a BSA between 20% and 39%, a Physician’s Global Assessment (PGA) 3, and a DLQI 30.

In June 2021, the patient started risankizumab and after 12 weeks of treatment returned for a follow-up evaluation with a significant improvement of the skin clinical picture (PASI 3,4 BSA between 3 and 11%, PGA 1, DLQI 3), as showed in Figure 13.

Key Points

Risankizumab showed consistent efficacy regardless of the localization and the characteristics of the disease with fast onset of action [14].

#### 2.1.8. Multi-Failure Patient

A 49-year-old male patient with psoriasis since about the age of 29 years came to our observation in December 2018, when the patient was under ustekinumab treatment with partial response. The patient referred to previous treatments with cyclosporine, infliximab, adalimumab, and then ustekinumab, which led to a partial disease control.

Objective examination revealed typical plaques in the face, trunk, and upper and lower limbs, PASI 15, DLQI 28, and BSA 24%.

In September 2019, the patient started the treatment with risankizumab (150 mg s.c.). In January 2020, after three administrations, the clinical scenario was almost solved, with a PASI 2.2, BSA 6%, and DLQI 6. The improvement can be seen in Figure 14.

After two years of treatment, the clinical scenario has been completely resolved, with a PASI 0, DLQI 0, and BSA 0%.

Key points

Biologically experienced patients (with various primary or secondary failures to therapy) represent a challenge because of the burden of disease; it becomes difficult to be adherent to therapy because they have been disappointed so many times and worried about recurrence of disease even if they present as clear. Long-term stable efficacy represents a key point in such patients, as well as fast onset of response.It is worth underlining the patient’s compliance to the treatment due to the convenient way of administration of risankizumab (every 3 months).

## 3. Discussion

Psoriasis is an immune-mediated chronic disease associated with a deep impairment in quality of life [15]. Moreover, the impact of the disease in patient functional, psychological, and social outcomes is highly influenced by the body area where the disease occurs [16].

According to the British Association of Dermatology, biological therapy should be started when psoriasis deeply impacts on physical, psychological, or social functioning (for example, Dermatology Life Quality Index (DLQI) or clinically relevant depressive or anxiety symptoms) and one or more of the following disease severity criteria apply: extensive psoriasis (defined as body surface area -BSA- > 10% or Psoriasis Area and Severity Index -PASI- ≥ 10) or severe psoriasis localized in sensitive areas and associated with significant functional impairment and/or high levels of distress (for example nail disease or involvement of high-impact and difficult-to-treat sites such as the face, scalp, palms, soles, flexures, and genitals) [17].

Interleukin (IL)-23 contributes to psoriasis by stimulating proliferation, differentiation, and maintenance of T helper 17 cells and innate immune cells, as well as the production of different proinflammatory cytokines, such as IL-17 [18,19]. Currently, various approved biologics are effective for the treatment of plaque psoriasis, although some of them are not able to guarantee a long term-maintained response [18].

Risankizumab is a humanized IgG1 monoclonal antibody against the subunit p19 of IL-23, which showed greater efficacy in patients with moderate to severe plaque psoriasis than ustekinumab, adalimumab, and secukinumab [7,9,18].

In the IMMerge (a phase III, randomized, open-label, efficacy–assessor-blinded clinical trial) that compared the efficacy and safety of risankizumab vs. secukinumab, a smaller difference in PASI 90 response between IL-23- and IL-17A-targeted treatments at week 16 followed by larger differences favoring the IL-23 inhibitor after 52 weeks of treatment was recorded. Such a favorable difference in the long term (52 weeks) for risankizumab seems to confirm that inhibiting IL-23 seems to be more target-specific, increasing the durable efficacy under an excellent safety profile [18].

Although we reported only eight clinical cases, they represent real-world clinical conditions and, as such, they provide a valuable adjunct to results obtained in the selected patients usually included in controlled clinical trials.

## 4. Conclusions

Risankizumab is a humanized IgG1 monoclonal antibody against the subunit p19 of IL-23, which showed greater efficacy in patients with moderate to severe plaque psoriasis than ustekinumab, adalimumab, and secukinumab. This case collection represents a window into a daily practice setting, since our cases are representative of real-life scenarios, where we usually find comorbid and non-adherent-to-treatment patients, with forms of disease that are normally excluded from clinical trials. As demonstrated in our cases, risankizumab rapidly improved clinical manifestations and relieved symptoms in patients with moderate to severe psoriasis, regardless of the presence of comorbidities or the location of the plaques, and without any safety signal. As expected, a prominent level of clinical improvement correlates to the remarkable amelioration in patients’ quality of life, often deeply compromised.

## Figures and Tables

**Figure 1 biomedicines-11-01769-f001:**
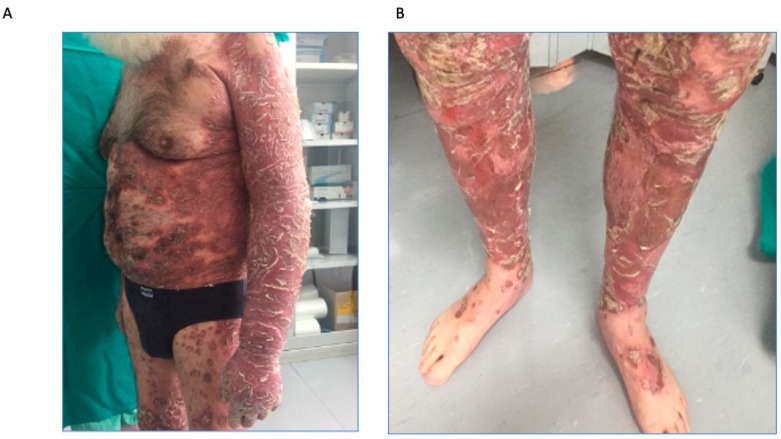
First objective examination in patient 1. (**A**), extensive infiltrated erythematous-squamous plaques; (**B**), marked involvement of the limbs.

**Figure 2 biomedicines-11-01769-f002:**
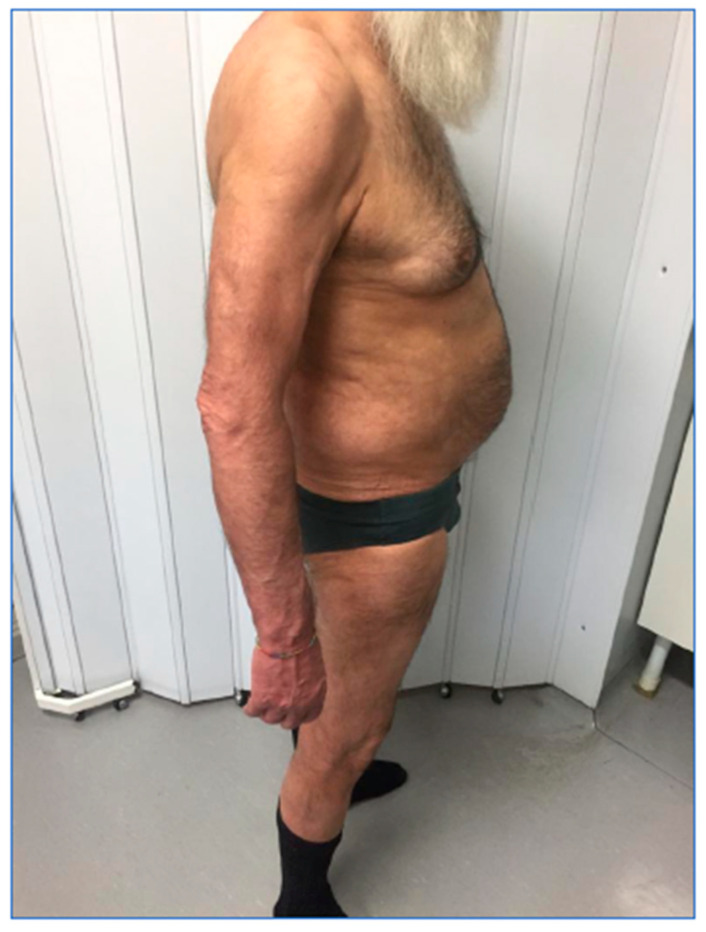
Patient during the follow-up visit.

**Figure 3 biomedicines-11-01769-f003:**
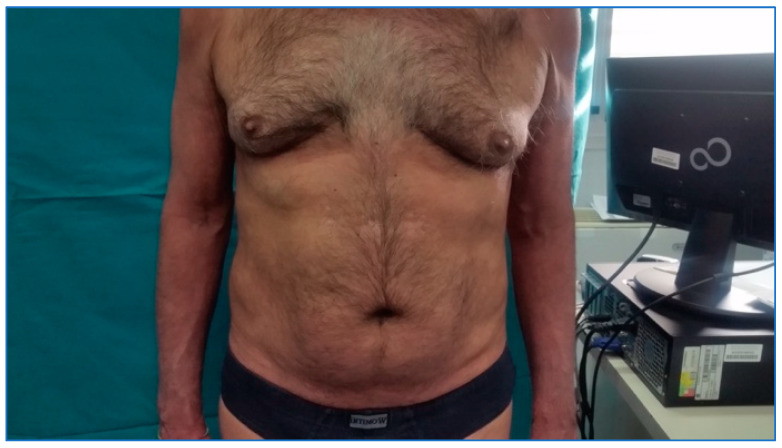
Patient evaluation after 24 months.

**Figure 4 biomedicines-11-01769-f004:**
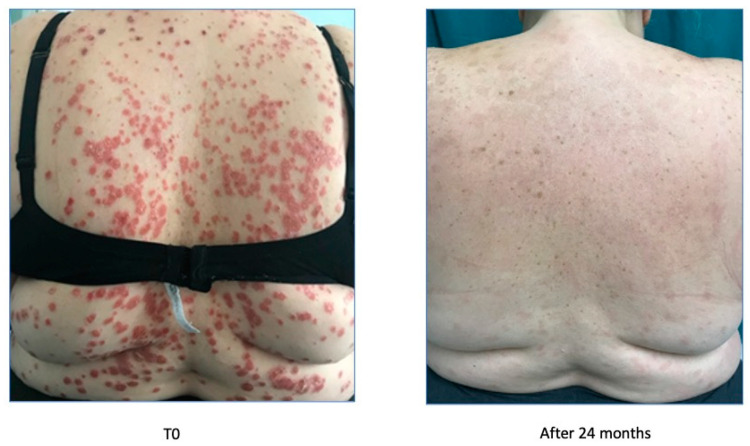
Psoriasis improvement after 24 months.

**Figure 5 biomedicines-11-01769-f005:**
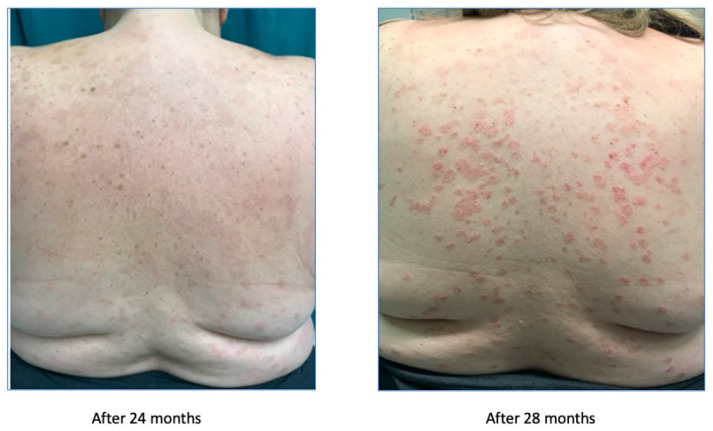
Treatment failure after 28 weeks.

**Figure 6 biomedicines-11-01769-f006:**
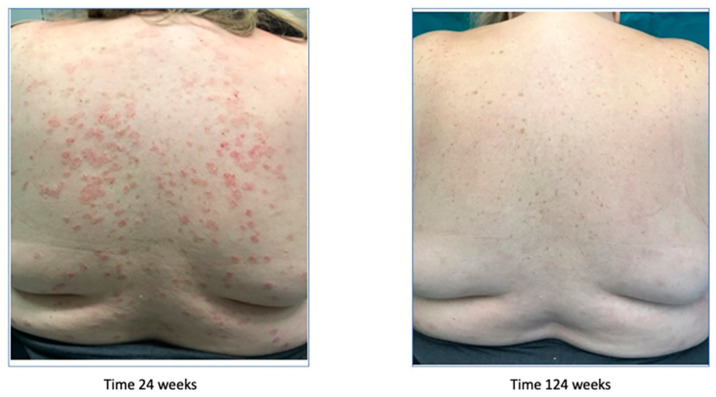
Psoriasis improvement after 124 weeks.

**Figure 7 biomedicines-11-01769-f007:**
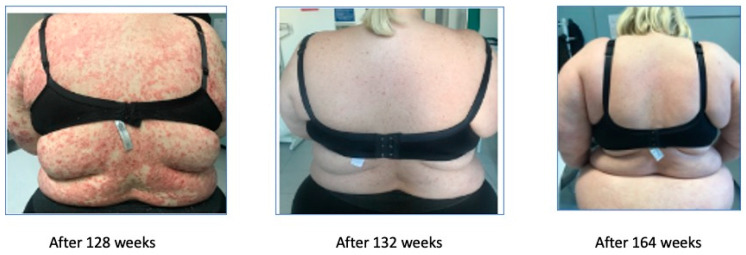
Complete remission after 164 weeks (3.4 years).

**Figure 8 biomedicines-11-01769-f008:**
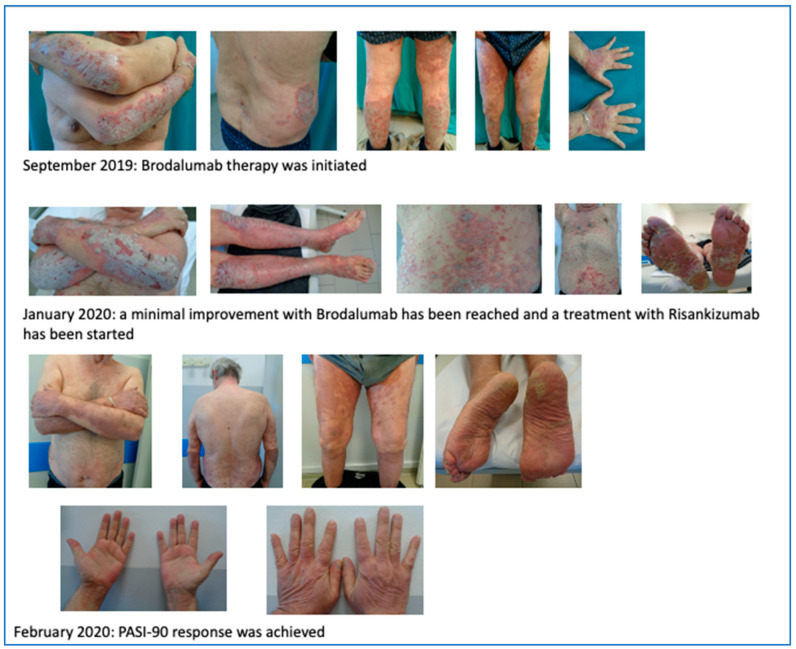
PASI-90 response after 4 weeks of treatment with risankizumab.

**Figure 9 biomedicines-11-01769-f009:**
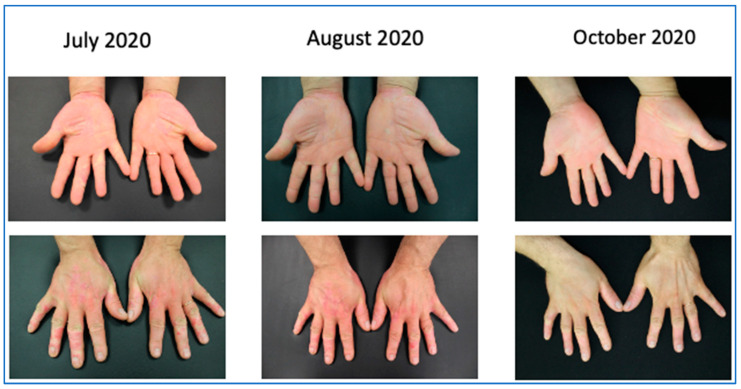
Complete remission of the symptoms in three months.

**Figure 10 biomedicines-11-01769-f010:**
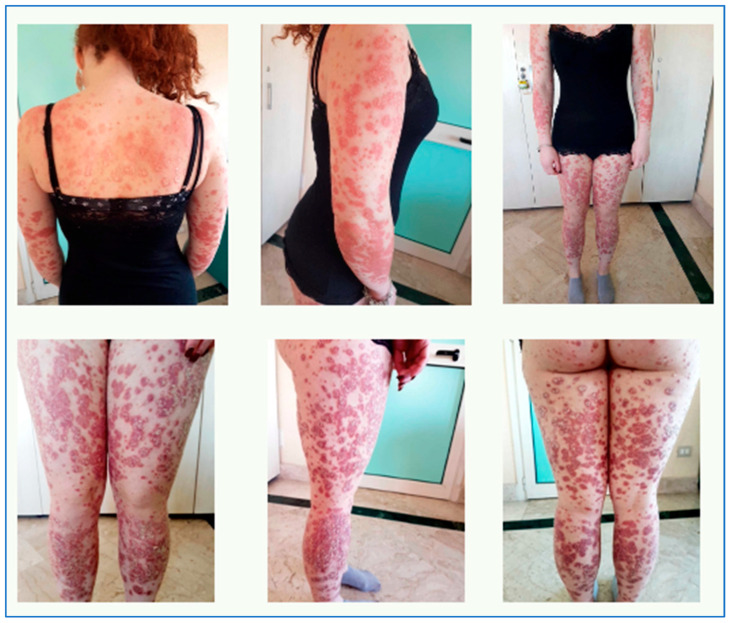
Objective examination at time 0.

**Figure 11 biomedicines-11-01769-f011:**
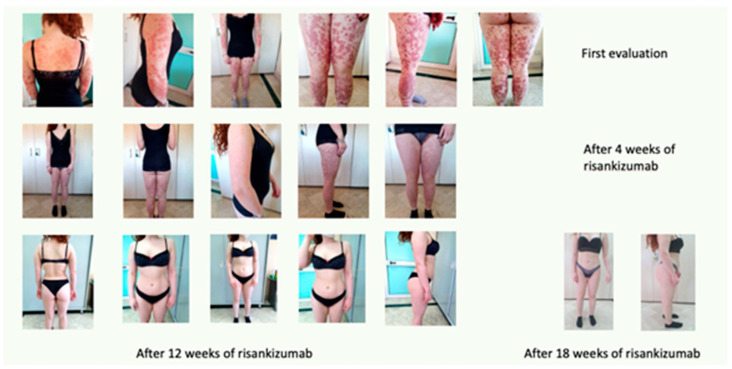
Complete remission after 18 weeks of treatment.

**Figure 12 biomedicines-11-01769-f012:**
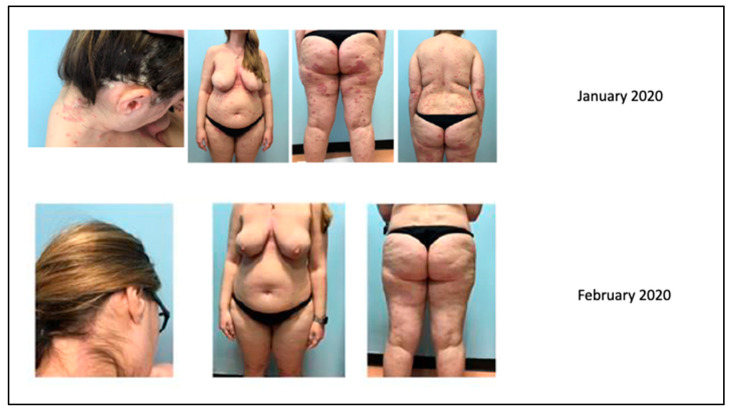
Complete remission after 1 month of treatment.

**Figure 13 biomedicines-11-01769-f013:**
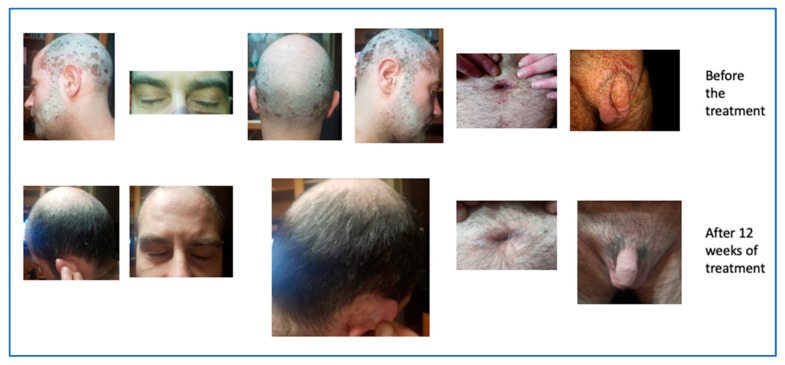
Complete remission after 12 weeks of treatment.

**Figure 14 biomedicines-11-01769-f014:**
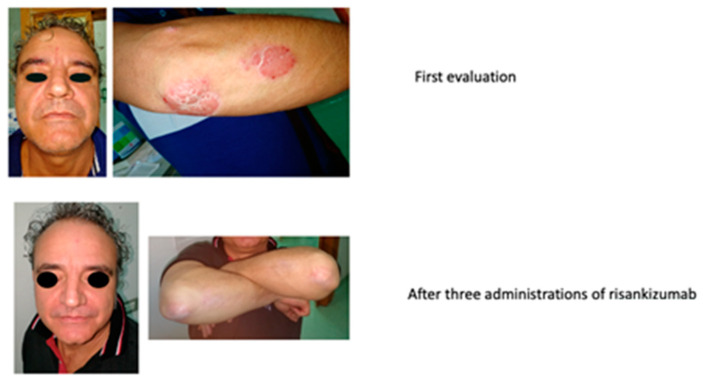
Complete remission after three administrations of risankizumab.

**Table 1 biomedicines-11-01769-t001:** Summary of patient’s characteristics, previous therapies, and reason for choosing therapy.

Patient Number (Sex, Age)	Comorbidities	Previous Therapies	Reason for Choosing Risankizumab
Patient 1 (M, 68 y)	Arterial hypertension, previous pulmonary tuberculosis, previous myocardial infarction with pacemaker	Contraindicated to systemic conventional therapies	Previous lack of adherence to therapy and comorbidities. Fast response and safety were major concerns.
Patient 2 (F, 45 y)	Obesity	Cyclosporine, ustekinumab, secukinumab, and ixekizumab	Long term response (due to secukinumab and ixekizumab loss of efficacy). Safety profile (due to adverse event: erythema multiforme under ustekinumab).
Patient 3 (M, 67 y)	Hypertension, hypercholesterolemia and previous heart infarct, previously resolved hepatitis B infection	Cyclosporine, systemic steroids, itraconazole, secukinumab, and brodalumab	Better short- and long-term response with risankizumab vs. anti IL-17 therapies. Excellent safety profile.
Patient 4 (M, 50 y)	Hepatitis B infection under treatment	Acitretin, etanercept, adalimumab, ustekinumab, secukinumab, and ixekizumab	Better short- and long-term response with risankizumab vs. anti IL-17 or anti TNF-a therapies. Excellent safety profile.
Patient 5 (F, 23 y)	None	Phototherapy, cyclosporine	Adherence to therapy as key factor and long-term maintenance of response.
Patient 6 (F, 38 y)	None	Cyclosporine	Efficacy for scalp psoriasis. Increase adherence to therapy.
Patient 7 (M, 46 y)	Hypercholesterolemia	Methotrexate, cyclosporine, acitretin, psoralen plus ultraviolet-A radiation (PUVA), adalimumab, and secukinumab	Excellent efficacy also in non-naive patients to biologicals. Long-term maintenance of response.
Patient 8 (M, 49 y)	None	Cyclosporine, infliximab, adalimumab, and ustekinumab	Excellent efficacy in biologically experienced patients. Long-term maintenance of response.

## Data Availability

Not applicable.

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
