# Peer review of "Risankizumab: Daily Practice Experience of High Need Patients"

_biomedicines, 2023, doi:10.3390/biomedicines11061769_

Round 1
Reviewer 1 Report
The authors should be consistent with the use of PASI score vs. PASI response. For example, “PASI 100” should be stated as “PASI-100 response” as with “PASI-90 response” to minimize confusion.
Line 52: the authors stated that the trials involved 994 subjects, of which, 506 in UltIMMa-1 and 491 in UltiMMa-2. Please clarify the subject counts.
Line 82: “regularly assumed alcohol…” should be “regularly consumed alcohol…”
Line 93: DLQI should be defined when initially described.
Line 100: “150 mg s.c in October” should be “150 mg s.c. on October”; the same inconsistency is observed in Line 287.
Line 120: is missing a “.” After “observation”
Line 230: “Adhecrence” is a typographical error
Line 323: the number of reported cases should be “eight” instead of “ten”
Discussion on the potential reasons why risankizumab was more effective than the other monoclonal Abs, i.e., ustekinumab, adalimumab, secukinumab, brodalumab, ixekizumab, etc., would increase the value of this manuscript.
A table summarizing the eight cases, e.g., listing the sex, age, comorbidities, and the initial mabs utilized for the initial treatment, as well as the rationals for choosing the cases, will improve the readability of the manuscript.
Minor errors have been identified as identified in the Comments section.
Author Response
Answers:
Reviewer No. 1:
"Line 52: the authors stated that the trials involved 994 subjects, of which, 506 in UltIMMa-1 and 491 in UltiMMa-2. Please clarify the subject counts." and Line 82: “regularly assumed alcohol…” should be “regularly consumed alcohol…”Thank you very much for pointing this out the total number of patients enrolled was charged to 997, see underlined text at line 52 and the word assumed was changed to consumed.
Line 93: DLQI should be defined when initially described nd Line 100: “150 mg s.c in October” should be “150 mg s.c. on October”; the same inconsistency is observed in Line 287. We appreciate the observation, (Dermatology Life Quality Index) was added in lines 102 and 103. Changes were made also from in to on in Line 100, 157 and 287.
Line 120: is missing a “.” After “observation”. It was added.
Line 230: “Adhecrence” is a typographical error. It was changed to "Adherence".
Line 323: the number of reported cases should be “eight” instead of “ten”. It was changed to "eight".
Discussion on the potential reasons why risankizumab was more effective than the other monoclonal Abs, i.e., ustekinumab, adalimumab, secukinumab, brodalumab, ixekizumab, etc., would increase the value of this manuscript. The appreciate the observation and the following text was added in Lines 323 to 327: "
In the IMMerge (a phase III, randomized, open‐label, efficacy–assessor‐blinded clinical trial) that compared the efficacy and safety of risankizumab vs. secukinumab, a smaller difference in PASI 90 response between IL–23– and IL‐17A‐targeted treatments at week 16 followed by larger differences favouring the IL‐23 inhibitor after 52 weeks of treatment was recorded. Such a favourable difference in the long term (52 weeks) for risankizumab seems to confirm that inhibiting IL-23 seems to be more target-specific increasing the durable efficacy under an excellent safety profile. (18)"
A table summarizing the eight cases, e.g., listing the sex, age, comorbidities, and the initial mabs utilized for the initial treatment, as well as the rationals for choosing the cases, will improve the readability of the manuscript. We provide a new table as suggested by the reviewer, see table 1. See underlined text on Lines 79-80 "
For a summary of patient’s characteristics, previous therapies and reason for choosing therapy seen Table 1."
Reviewer 2 Report
Alexandra M. G. Brunasso and colleagues present a quality and well-written case report focused on Risankizumab with regards to daily practice experience of high need patients.
Authors present 8 cases of patients with moderate-to-severe psoriasis treated with risankizumab with a significant efficacy in the remission of the disease. Their cases represent real-world clinical setting and provide a valuable adjunct to results obtained in the selected patients usually included in controlled clinical trials. In these cases, Risankizumab rapidly improved clinical manifestations and relieved symptoms in patients with moderate-to-severe psoriasis, regardless the presence of comorbidities or the location of the plaques in special sites, and without any safety concerns.
Authors suggest that although they reported only ten clinical cases, they represent real-world clinical conditions and, as such, they provide a valuable adjunct to results obtained in the selected patients usually included in controlled clinical trials.
Finally, authors conclude that this case collection represents a window to a daily practice setting, since their cases are representative of real-life scenario, where they usually find comorbid and non-adherent to treatment-patients, with forms of disease that are normally excluded from clinical trials. As demonstrated in their cases, risankizumab rapidly improved clinical manifestations and relieved symptoms in patients with moderate-to-severe psoriasis, regardless the presence of comorbidities or the location of the plaques, and without any safety signal. As expected, a prominent level of clinical improvement correlates to the remarkable amelioration in patients’s quality of life, often deeply compromised.
Overall, the manuscript is valuable for the scientific community and should be accepted for publication after edits are made.
===========================
Other comments:
1) Please check for typos throughout the manuscript.
2) Authors are kindly encouraged to cite the following article that describes novel targets in autoimmune diseases, of which psoriasis is a prominent example. DOI: 10.1007/s12668-016-0233-x
Author Response
1) Please check for typos throughout the manuscript. WE thank the reviewer for the suggestion, underlined changes have been made in the manuscript
2) Authors are kindly encouraged to cite the following article that describes novel targets in autoimmune diseases, of which psoriasis is a prominent example. DOI: 10.1007/s12668-016-0233-x. We appreciate the suggestione, with the DOI number we were able to find the following article, but we are not sure is it is the correct reference:
"Ubiquitin-proteasome system: Promising therapeutic targets in autoimmune and neurodegenerative diseases
E Bulatov, S Khaiboullina, HJ dos Reis, A Palotás… - BioNanoScience, 2016 - Springer"Reviewer 3 Report
Dear Authors,
I read your manuscript concerning the use of Risankizumab in clinical practice and high-need patients. The paper is a case series and reports a peculiar setting where you can use an IL-23 inhibitor. Some points must be clarified.
1) High-need patients, explain and clarify.
2) Line 37, “intensive treatment”, explain.
3) Lines 26,43, risankizumab, please.
4) Lines 48-49, moderate-to-severe, please.
5) Line 77. Report national guidelines and government notices.
6) Line 86, specify topical treatments.
7) Lines 95-103, format.
8) In the reported cases, did you evaluate blood profiles? Significant differences?
9) Case 2. Definition of obesity and WHO classification for the patient. Line 119, BMI unit is missing. I think the case concerns metabolic syndrome. Can you check?
10) Case 3. Did you suspect a fungal infection? Itraconazole in the anamnesis? Format final lines.
11) Line 237, correct.
12) Lines 261,262 report anamnesis details and blood exams.
13) No itch VAS, PGA or other specific scores have been evaluated in the single cases. Update this information if presented.
14) Update the informed consent statement with government notices.
15) Study limitation section is missing.
16) Line 301, fullstop.
17) It should be helpful for the reader to have a resume (table) that reports all the cases and clinical aspects.
18) You should read and cite the following articles to improve the discussion and clinical approach for each case:
- doi: 10.3390/ph16040526
- doi: 10.1111/jcmm.15742
Minor English language editing is required, but an extensive check of the form must be performed.
Author Response
We thank the reviewer for all the comments and herein we provide an answer.
1) High-need patients, explain and clarify. Please see underlined text in Lines 38 to 42 "
High-need patients have been defined in the present case series as adult cases with moderate to severe plaque psoriasis who were either not controlled by, or were intolerant to or had contraindications to at least two currently available systemic therapies (eg, photochemotherapy, cyclosporine, methotrexate, oral retinoids, fumaric acid esters and biologicals) o require due to high severity a fast control of the disease."
2) Line 37, “intensive treatment”, explain. WE changed the word intensive with "effective"
3) Lines 26,43, risankizumab, please. See underlined changes in lines 27 and 46.
4) Lines 48-49, moderate-to-severe, please. Changes were made.
5) Line 77. Report national guidelines and government notices. See underlined tex in Lines 75-78.
6) Line 86, specify topical treatments. See underlined text in Line 97 "
corticosteroids and vitamin D derivates"7) Lines 95-103, format. The format was corrected.
8) In the reported cases, did you evaluate blood profiles? Significant differences? No blood profiles were evaluated because they are not routinely performed in the every day practice.
9) Case 2. Definition of obesity and WHO classification for the patient. Line 119, BMI unit is missing. I think the case concerns metabolic syndrome. Can you check? BMI is present in line 128.
10) Case 3. Did you suspect a fungal infection? Itraconazole in the anamnesis? Format final lines. Itraconazole was prescribed by a private dermatologist prior to our visit and we don't know if fungal infection was suspected.
11) Line 237, correct. See correction in Line 246.
12) Lines 261,262 report anamnesis details and blood exams. Anamnesis details are present in Lines 270 to 276.
13) No itch VAS, PGA or other specific scores have been evaluated in the single cases. Update this information if presented. PASI and DLQI scores were evaluated and in some cases also PGA was assessed, Itch VAS was not recorded as in most clinical trials regarding psoriasis.
14) Update the informed consent statement with government notices. see the new attached document.
15) Study limitation section is missing. See Lines 346-348.
16) Line 301, fullstop.
17) It should be helpful for the reader to have a resume (table) that reports all the cases and clinical aspects. See table 1 added to the manuscript.
18) You should read and cite the following articles to improve the discussion and clinical approach for each case: we thank the reviewer ref 19 was added, see underlined text in Line 417-418
- doi: 10.3390/ph16040526
- doi: 10.1111/jcmm.15742
Round 2
Reviewer 3 Report
Dear Authors,
all the corrections have been made and the manuscript has been improved.